# Association of the *CFTR* gene with asthma and airway mucus hypersecretion

**Astrid Crespo-Lessmann**[1]*, **Sara Bernal**[2], **Elisabeth del Río**[2], **Ester Rojas**[2], **Carlos Martínez-Rivera**[3], **Nuria Marina**[4], **Abel Pallarés-Sanmartín**[5], **Silvia Pascual**[6], **Juan Luis García-Rivero**[7], **Alicia Padilla-Galo**[8], **Elena Curto**[1], **Carolina Cisneros**[9], **José Serrano**[10], **Montserrat Baiget**[2], **Vicente Plaza**[1], **Emerging Asthma Group**[¶]

**1** Department of Respiratory Medicine, Hospital de la Santa Creu i Sant Pau, Institute of Sant Pau Biomedical Research (IBB Sant Pau), Centro de Investigación Biomédica en Red de Enfermedades Respiratorias (CIBERES), Universitat Autònoma de Barcelona, Barcelona, Spain, **2** Department of Genetics, Hospital de la Santa Creu i Sant Pau, Institute of Sant Pau Biomedical Research (IBB Sant Pau), Centro de Investigación Biomédica en Red de Enfermedades Raras (CIBERER, U705), Barcelona, Spain, **3** Department of Respiratory Medicine, H. German Trias i Pujol, CIBERES, Badalona, Spain, **4** Department of Respiratory Medicine, H. de Cruces, Barakaldo, Vizcaya, Spain, **5** Department of Respiratory Medicine, H. Álvaro Cunqueiro, Vigo, Spain, **6** Department of Respiratory Medicine, H. de Galdakao, Vizcaya, Spain, **7** Department of Respiratory Medicine, H. Laredo, Cantabria, Spain, **8** Department of Respiratory Medicine, H. Costa del Sol de Marbella, Málaga, Spain, **9** Department of Respiratory Medicine, H. U. de La Princesa, Madrid, Spain, **10** Department of Respiratory Medicine, Hospital Comarcal de Inca, Baleares, Spain

¶ Membership of the Emerging Asthma Group is Provided in the Acknowledgments.
* acrespo@santpau.cat

**Data Availability Statement:** All relevant data are within the paper and its Supporting Information files.

## Abstract

### Introduction

Asthma with airway mucus hypersecretion is an inadequately characterized variant of asthma. While several studies have reported that hypersecreting patients may carry genetic variants in the cystic fibrosis transmembrane conductance regulator (*CFTR*) gene, many of those studies have been questioned for their numerous limitations and contradictory results.

### Objectives

(1) To determine the presence of genetic variants of the *CFTR* gene in patients with asthma with and without airway mucus hypersecretion. (2) To identify the clinical, inflammatory and functional characteristics of the asthma phenotype with airway mucus hypersecretion.

### Method

Comparative multicentre cross-sectional descriptive study that included 100 patients with asthma (39 hypersecretors and 61 non-hypersecretors). Asthmatic hypersecretion was defined as the presence of cough productive of sputum on most days for at least 3 months in 2 successive years. The patients were tested for fractional exhaled nitric oxide, spirometry, induced sputum cell count, total immunoglobulin E (IgE), peripheral blood eosinophil count, C-reactive protein, blood fibrinogen and blood albumin and underwent a skin prick test. Asthma control and quality of life were assessed by the Asthma Control Test and Mini Asthma Quality of Life questionnaires, respectively. Blood DNA samples were collected

**Funding:** Grant from de Spanish Society of Pulmonology and Thoracic Surgery (SEPAR) to Dr. Astrid Crespo Lessmann, and Astra Zeneca. The funders had no role in study design, data collection and analysis, decision to publish, or preparation of the manuscript.

**Competing interests:** I have read the journal's policy and the authors of this manuscript have the following competing interests: AC has received fees in the last three years for talks at meetings sponsored by Chiesi, Esteve Laboratories, GlaxoSmithKline, Novartis, Ferrer, Zambón and Boehringer Ingelheim, has received travel and attendance expenses for conferences from Novartis, Bial, Teva and FAES Farma and has received funds/grants for research projects from several state agencies and non-profit foundations and from AstraZeneca. EC reports non-financial support from Astrazeneca, personal fees from Boehringer-Ingleheim, personal fees and non-financial support from Chiesi, non-financial support from Novartis, non-financial support from Menarini, non-financial support from ALK, outside the submitted work. SB, EdR, ER, CM, NM, AP, SP, JG, AP, CC, JS and MB have no conflicts of interest to declare. VP has received fees in the last three years for talks at meetings sponsored by AstraZeneca, Boehringer-Ingelheim, MSD and Chiesi, has received travel and attendance expenses for conferences from AstraZeneca, Chiesi and Novartis, has acted as a consultant for ALK, AstraZeneca, Boehringer, MSD, MundiPharma and Sanofi, and has received funds/grants for research projects from several state agencies and non-profit foundations and from AstraZeneca, Chiesi and Menarini. This does not alter our adherence to PLOS ONE policies on sharing data and materials.

from the patients and next-generation sequencing using a MiSeq sequencer and the Illumina platform was used for the *CFTR* gene analysis.

## Results

Genetic differences were observed in the c.1680-870T>A polymorphism of the *CFTR* gene, significantly more evident in hypersecretors than in non-hypersecretors: 78.94% vs. 59.32% in the majority allele and 21.05% vs. 40.67% in the minority allele (p = 0.036). Clinically, asthma hypersecretors compared to non-hypersecretors were older (57.4 years vs. 49.4 years; p = 0.004); had greater asthma severity (58.9% vs. 23.7%; p = 0.005); experienced greater airway obstruction (FEV1/FVC% 64.3 vs. 69.5; p = 0.041); had poorer asthma control (60% vs. 29%; p = 0.021); had lower IgE levels (126.4 IU/mL vs. 407.6 IU/mL; p = 0.003); and were less likely to have a positive prick test (37.5% vs. 68.85%; p = 0.011).

## Conclusion

The results suggest that patients with asthma and with mucus hypersecretion (1) may have a different phenotype and disease mechanism produced by an intronic polymorphism in the *CFTR* gene (NM_000492.3:c.1680-870T>A), and (2) may have a poorer clinical outcome characterized by severe disease and poorer asthma control with a non-allergic inflammatory phenotype.

## Introduction

In clinical practice, it is common to encounter patients with severe asthma who only partially respond to steroid treatment and who have marked airway mucus hypersecretion. The increase in mucus secretion may be mild or severe, marked only by chronic bronchitis or by recurring bronchial infections, respectively; in extreme cases, patients may develop infectious bronchiectasis and bronchiolitis. Probably because 'airway mucus secretion' is not usually included in cluster analyses of asthma, this condition is not recognized as a specific phenotype.

Wheaterall et al [1] evaluated patients with airflow obstruction and airway mucus hypersecretion, finding a poorer response to glucocorticoid treatment and more frequent hospital admissions due to exacerbations in patients with airway mucus hypersecretion secondary to chronic obstructive pulmonary disease (COPD) or asthma (bronchial hyperresponsiveness, elevated fractional exhaled nitric oxide (FeNO), rhinitis, dermatitis and blood eosinophilia). Mutations or polymorphisms in the cystic fibrosis transmembrane conductance regulator (*CFTR*) gene were detected in another study of 4 hypersecreting patients with asthma, with a neutrophilic inflammatory phenotype, bronchiectasis, pansinusitis and respiratory infections [2]; this would suggest that a possible explanation for airway mucus hypersecretion may be that having asthma results in a combination of asthma and an attenuated form of cystic fibrosis (CF) in patients with a genetic alteration in the *CFTR* gene. By the 1990s, several studies had confirmed that carriers of a mutation for CF were at an increased risk of asthma and of greater lung function deterioration than patients with asthma without that mutation [3, 4]. However, several works published between 1998 and 2008 have been questioned since, on the basis that they report contradictory results [3–6]; furthermore, they had limitations that included incorrectly diagnosing asthma, using emergency registry databases, detecting just a single *CFTR* gene mutation and analysing different populations with unequal comparisons in terms of sex and age [5, 6]. Furthermore, all types of asthma were included, i.e., no phenotype filtering was applied.

CF is the most common severe autosomal recessive hereditary disease reported for Caucasians; in Spain incidence is estimated as 1 in 5,352 full-term births [7]. CF is caused by mutations in the *CFTR* gene residing in the long arm of human chromosome 7 and coding for the CFTR protein [8, 9], in turn, acting as a chloride channel on the surface of epithelial cells in a wide variety of organs (thereby explaining the wide range of clinical manifestations of CF).

To date, of some 1,900 genetic variants described for the *CFTR* gene, most frequent in the Caucasian population is the three-base-pair deletion NM_000492.3(CFTR): c.1521_1523delCTT (p.Phe508del) or F508del according to the classical nomenclature [10, 11]. Since the inheritance pattern for CF is autosomal recessive, both copies of the mutated *CFTR* gene must be present for the disease to develop, whereas only a single copy is necessary to be a CF carrier. It is estimated that around 4%-5% of the general population are CF carriers [12].

The main line of work in our current project is the search for genetic variants (mutations or polymorphisms) in the *CFTR* gene using mass sequencing platforms and next-generation sequencing (NGS) and, more specifically, the investigation of airway mucus hypersecretors and non-hypersecretors with asthma and pre-defined clinical characteristics. The goal is, by considering the phenotype and genotype of the *CFTR* gene, to achieve better stratification in asthma classifications and so improve disease prognosis for hypersecreting patients and, ultimately, enhance the efficiency of current treatments and develop targeted and personalized future treatments.

## Materials and methods

Comparative multicentre cross-sectional descriptive study, designed to determine the presence of genetic variants (mutations or polymorphisms) of the *CFTR* gene in hypersecreting and non-hypersecreting patients with asthma and to identify the clinical and inflammatory characteristics of hypersecreting patients.

Included were 100 patients of both sexes, aged between 18 and 80 years, with and without airway mucus hypersecretion; all of them complied with asthma diagnostic criteria according to the Global Initiative for Asthma (GINA) guidelines published in 2009 [13] and none of them had a respiratory infection in the month before testing. Excluded were smokers and ex-smokers, patients with lung conditions (tuberculosis sequelae, bronchiectasis, CF, residual pleural diseases, interstitial diseases and severe associated comorbidities) and patients treated with oral corticosteroids or other immunomodulators for reasons other than asthma. Included patients were administered a questionnaire to evaluate symptoms and type of expectoration (S1 Appendix). Asthmatic hypersecretion was defined as the presence of cough productive of sputum on most days for at least 3 months in 2 successive years.

### Ethical considerations

The study was conducted according to the principles of the Declaration of Helsinki (18th World Medical Assembly) and was approved by the Hospital de la Santa Creu i Sant Pau (HSCSP, Barcelona) Clinical Research Ethics Committee (approval number COD: HBSP-CFT-2014-68; ClinicalTrials.gov Identifier: NCT02558127). All patients signed an informed consent prior to participating in the study and were guaranteed the confidentiality of their data.

### Method

Included between 2014 and 2016 were 100 patients with asthma (39 hypersecretors and 61 non-hypersecretors), 98 of whom were analysed (2 withdrew from the study). The patients

included in the study, who came from the outpatient clinics of the hospitals participating in the study, were recruited according to the criteria of the participating physicians. Exclusion criteria were the presence of other associated lung conditions (allergic bronchopulmonary aspergillosis, tuberculosis sequelae, severe bronchiectasis secondary to a respiratory disease other than asthma, CF, residual pleural diseases and interstitial diseases); severe associated comorbidities; and treatment with oral corticosteroids or other immunomodulators for causes other than asthma.

Tests were performed as follows: fractional exhaled nitric oxide (FeNO), spirometry, induced sputum cell count (conducted only at the HSCSP), total immunoglobulin E (IgE), peripheral blood eosinophil count, C-reactive protein, blood fibrinogen, blood albumin and skin prick test for common pneumoallergens. All patients completed the validated Spanish version of the Asthma Control Test (ACT) questionnaire [14] and the Mini Asthma Quality of Life (Mini-AQLQ) questionnaire [15]. The sweat chloride test could not be performed, first because we do not have the necessary equipment for adults, and second, because when carrying out a mass sequencing genetic study, if the patient had CF, this would have been detected.

FeNO was measured before spirometry using electrochemical equipment (NO Vario Analyzer, FILT Lungen and Thorax Diagnostic GmbH, Berlin, Germany) on the basis of expiration yielding a continuous flow of 50 mL/s of total lung capacity in accordance with American Thoracic Society (ATS)/European Respiratory Society (ERS) recommendations from 2005 [16]. A high FeNO value was defined as ≥50 parts per billion (ppb) [17]. Spirometry was performed with a Datospir-600 spirometer (Sibelmed SA, Barcelona, Spain) by an experienced pulmonary function technician who applied Spanish Society of Pulmonology and Thoracic Surgery (SEPAR) recommendations [18, 19]. Induced sputum samples were collected, according to the ERS consensus protocol [20], using an ultrasonic nebulizer (Omron NE U07, HEALTHCARE Europe, Germany)–output 3 mL/s and mass mean aerodynamic diameter (MMAD) particle size 7 μm–to produce a hypertonic saline spray which the patient inhaled at increasing concentrations (3%, 4% and 5%). Sputum was processed within 2 hours of induction. Patients were classified as having neutrophilic asthma when neutrophil count was ≥61%, eosinophilic asthma when eosinophil count was ≥3%, paucigranulocytic asthma when neutrophil count was <61% and eosinophil count was <3%, and mixed asthma when neutrophil count was ≥61% and eosinophil count was ≥3% [21]. Total serum IgE was measured by ImmunoCAP using the UniCAP 250 system (Phadia AB, Uppsala, Sweden), with a high value defined as >160 IU/mL. The prick test, performed as standard, was considered positive for wheal diameters >3 mm [22]. Controlled asthma was defined as an ACT score ≥20. A high dose of inhaled glucocorticoids was defined as ≥1,000 mg/day of beclomethasone dipropionate or equivalent [23].

## Study of genetic variants in the *CFTR* gene

A 10-mL peripheral venous blood sample was extracted by the saline method from patients for analysis by the genetics service of the reference hospital. DNA was analysed for genetic variants of the *CFTR* gene by means of NGS using the Illumina platform MiSeq sequencer and predesigned Multiplicom *CFTR* gene panels (CFTR Master Dx kit) as diagnostic tests, validated by a CE-IVD certificate and compatible with Illumina NGS sequencers. The NGS workflow consisted of 5 main steps: (1) library preparation using the CFTR Master Dx and MID Dx kits specific to the Illumina MiSeq for sample analyses; (2) cluster generation; (3) sequencing; (4) data analysis with the MiSeq Reporter software pre-installed in the MiSeq sequencer; and (5) data interpretation using computerized systems and databases incorporated in analytical software. Alignment and variant calling were implemented using standard methodologies. Briefly,

the reads were aligned against GRCh37 human genome assembly using Burrows-Wheeler Aligner (BWA) software. Variant calling was performed using the Genome Analysis Toolkit (GATK). Variant quality filtering parameters included a minimum of 30 reads, with at least 30% with the variant to call heterozygous genotypes. To prioritize variants, minor allele frequency (MAF) in the Genome Aggregation Database <0.05 was applied. Common variants were excluded by investigating the dbSNP, 1000G, ExAc and GnomAD v.2.1 databases. Mean vertical coverage was 1046X and mean horizontal coverage was 98.8% (Q≥30). Steps 2–4 were performed automatically by the MiSeq sequencer. The genetic variants identified using NGS were validated by Sanger sequencing. In parallel, possible complete or partial *CFTR* gene deletions and/or duplications were identified using Multiplex Ligated Probe Amplification (MLPA).

## Statistical analysis

Descriptive baseline values are reported as frequencies and percentages for qualitative data and means and standard deviation (SD) for quantitative data. The 2 asthma groups (hypersecretors and non-hypersecretors) were compared in terms of means and SD using the student-t test for the main variable and the remaining quantitative variables. The categorical variables, described by means of contingency tables and tested for differences using the chi-square test, are reported as numbers and percentages. Statistical significance was set to 5% (α = 0.05). Given the exploratory nature of the study, no correction was applied to multiple comparisons for over-dimensioning of the type I error. Statistical analyses were performed using SPSS (version 19.0) for Windows (SPSS, Inc., Chicago, Il, USA).

## Results

### Genetic results

Of the 98 DNA samples obtained from the peripheral blood of patients with asthma, 97 were analysed and 1 was excluded as being unsuitable for processing. A total of 50 genetic variants were found in the *CFTR* gene in 86 patients (the remaining 12 patients did not show any of these genetic variants: 11 were non-hypersecretors and the remaining case had an unknown phenotype). Significant differences were only observed between the 2 groups in a single genetic variant, namely NM_000492.3 (CFTR): c.1680-870T>A, present in hypersecretors but not in non-hypersecretors: 78.94% vs. 59.32% in the majority allele (thymine nucleotide), and 21.05% vs. 40.67% in the minor allele (adenine nucleotide) (p = 0.036) (see Table 1). Several samples had more than a single variant of the *CFTR* gene (86 patients). Two patients, one of whom was patient #4 in Table 2, showed 2 changes in the *CFTR* gene; the other patient had 2 changes in heterozygosis (c.220C>T and c.3808G>A), both predicting amino acid changes–p.(Arg74Trp) and p.(Asp1270Asn)–in the protein encoded by the *CFTR* gene. The clinical effect associated with both genetic variants is defined as having varying clinical consequences– 15% and 17% pancreatic insufficiency, respectively–according to the CFTR2 database (2020).

Of the 97 samples analysed, therefore, 4 showed pathological mutations (Table 2) already described in the literature and associated with CF. As all these mutations were detected in heterozygosis, patients who were carriers of the disease were informed. This finding confirms the internal validity of our results, since the proportion coincides with the 4%-5% risk of being a CF carrier described in the literature for the Caucasian population [12].

As a secondary objective of the study, polymorphisms were analysed in relation to asthma severity, with significant differences observed for 2 polymorphisms, namely, NM_000492.3 (CFTR): c.2506G>T (p.Asp836Tyr) and NM_000492.4(CFTR): c.3140-92T>C (p = 0.024 and

**Table 1. Main polymorphisms associated with the *CFTR* gene in patients with asthma with and without airway mucus hypersecretion.**

| Genetic variant | Hypersecretors (N = 38) | Non-hypersecretors (N = 59) | p |
|---|---|---|---|
| c.1680-870T>A | **21,05%/78,94%** | **40,67%/59,32%** | **0.036** |
| | **(N = 8/30)** | **(N = 24/35)** | |
| c.1680-871A>G | 100%/0% | 98.3%/1.69% | 0.608 |
| | (N = 38/0) | (N = 58/1) | |
| c.1727G>C [p.(Gly576Ala)] | 94.73%/5.26% | 98.3%/1.69% | 0.339 |
| | (N = 36/2) | (N = 58/1) | |
| c.2002C>T [p.(Arg668Cys)] | 92.10%/7.89% | 98.3%/1.69% | 0.165 |
| | (N = 35/3) | (N = 58/1) | |
| c.2047_2052delAAAAAAinsAAAAG [p.(Lys684Serfs*38)] | 97.36%/2.63% | 100%/ | 0.392 |
| | (N = 37/1) | (N = 59) | |
| c.2260G>A [p.(Val754Met)] | 97.36%/2.63% | 100%/ | 0.392 |
| | (N = 37/1) | (N = 59) | |
| c.2506G>T [p.(Asp836Tyr)] | 97.36%/2.63% | 94.91%/5.08% | 0.488 |
| | (N = 37/1) | (N = 56/3) | |
| c.2562T>G (p. =) | 50%/50% | 54.23%/45.76% | 0.421 |
| | (N = 19/19) | (N = 32/27) | |
| c.2619+3A>G | 100%/ | 98.3%/1.69% | 0.608 |
| | (N = 38/0) | (N = 58/1) | |
| c.2619+85_2619+86delAT | 44.73%/55.26% | 54.23%/45.76% | 0.24 |
| | (N = 17/21) | (N = 32/27) | |
| c.2619+106T>A | 86.84%/13.55% | 86.44%/13.55% | 0.604 |
| | (N = 33/5) | (N = 51/8) | |
| c.2909-71G>C | 97.36%/2.63% | 91.52%/8.47% | 0.238 |
| | (N = 37/1) | (N = 54/5) | |
| c.2909-92G>A | 65.78%/34.21% | 71.18%/28.81% | 0.366 |
| | (N = 25/13) | (N = 42/17) | |
| c.2991G>C [p.(Leu997Phe)] | 100%/ | 98.3%/1.69% | 0.608 |
| | (N = 38) | (N = 58/1) | |
| c.3139+42A>T | 100%/ | 98.3%/1.69% | 0.608 |
| | (N = 38) | (N = 58/1) | |
| c.3140-92T>C | 97.36%/2.63% | 89.83%/10.16% | 0.16 |
| | (N = 37/1) | (N = 53/6) | |
| c.3285A>T [p.(Thr1095Thr)] | 97.36%/2.63% | 100%/ | 0.392 |
| | (N = 37/1) | (N = 59) | |
| c.3367+37G>A | 97.36%/2.63% | 100%/ | 0.392 |
| | (N = 37/1) | (N = 59) | |
| c.3705T>G [p.(Ser1235Arg)] | 92.10%/7.89% | 96.61%/3.38% | 0.299 |
| | (N = 35/3) | (N = 57/2) | |
| c.3808G>A [p.(Asp1270Asn)] | 100%/ | 98.3%/1.69% | 0.808 |
| | (N = 38) | (N = 58/1) | |
| c.3870A>G | 97.36%/2.63% | 91.52%/8.47% | 0.238 |
| | (N = 37/1) | (N = 54/5) | |
| c.3874-200G>A | 94.74%/5.26% | 94.91%/5.08% | 0.654 |
| | (N = 36/2) | (N = 56/3) | |
| c.3909C>G [p.(Asn1303Lys)] | 100%/ | 98.3%/1.69% | 0.608 |
| | (N = 38) | (N = 58/1) | |

(*Continued*)

**Table 1.** (Continued)

| Genetic variant | Hypersecretors (N = 38) | Non-hypersecretors (N = 59) | p |
|---|---|---|---|
| **c.4137-139G>A** | 57.89%/42.10% | 71.18%/28.81% | 0.13 |
| | (N = 22/16) | (N = 42/17) | |
| **c.4243-20A>G** | 100%/ | 98.3%/1.69% | 0.608 |
| | (N = 38) | (N = 58/1) | |

CFTR: cystic fibrosis transmembrane conductance regulator.

p = 0.049, respectively), more frequently present in majority alleles in patients with severe persistent asthma, and absent or present to a lesser extent in minority alleles (Table 3).

## Clinical hypersecretor and non-hypersecretor phenotypes

The clinical, inflammatory and functional characteristics of hypersecretor and non-hypersecretor patients with asthma are shown in Table 4; of the 98 patients analysed, 39 were hypersecretors and 59 were non-hypersecretors. To a significant degree (p<0.05), hypersecretors compared to non-hypersecretors were older, had severer asthma, experienced greater bronchial obstruction, had poorer asthma control (ACT <20), had received more short-term oral glucocorticoid treatments in the previous year, had lower peripheral blood albumin levels, induced sputum lymphocyte levels and IgE levels, and were less likely to have prick test-positive asthma.

## Discussion

Our study reflects that asthma associated with airway mucus hypersecretion may be linked to an intronic *CFTR* gene polymorphism (NM_000492.3 (CFTR): c.1680-870T>A). It should be noted, however, that the sample used for this study may not be representative, so this association, despite being significant, cannot be considered to be a definite linkage. Nonetheless, the patients with this mutation–older, with more severe asthma and with poorer clinical control–represented a non-allergic predominantly eosinophilic inflammatory phenotype.

Airway mucus hypersecretion is a frequent symptom in patients with asthma, but unlike COPD, it is not usually taken into account in clinical practice or in clinical trials. Several published works have demonstrated that airway mucus hypersecretion is associated with greater asthma severity, is a marker of reduced lung function in both smokers and non-smokers, is associated with an increased number of exacerbation episodes and is a predictor of a poorer response to anti-inflammatory treatment with glucocorticoids [24–26]. Interestingly, in a

**Table 2. Pathological mutations described in the analysed population with asthma.**

| Patient No. | Mutation | Exon |
|---|---|---|
| 1 | c.2047_2052delAAAAAAinsAAAAG [p.(Lys684Serfs*38)] | E14 |
| 2 | c.3909C>G [p.(Asn1303Lys)] | E24 |
| 3 | c.1521_1523delCTT [p.(Phe508del)] | E11 |
| 4 | c.350G>A [p.(Arg117His)]; c.1521_1523delCTT [p.(Phe508del)] | E4; E11 |

Patient #4 presented with 2 heterozygous changes: pathological (c.1521_1523delCTT) and change c.350G>A, the latter classified as missense since an amino acid change is predicted, although it is described in the cystic fibrosis database as a genetic variant with variable clinical consequences.

The 4 patients with cystic fibrosis presented with a pathological mutation in heterozygosis.

**Table 3. Polymorphisms associated with the *CFTR* gene according to asthma severity.**

| GENOTYPE | Intermittent asthma | Persistent asthma | | |
|---|---|---|---|---|
| | | Mild | Moderate | Severe |
| | (N = 25) | (N = 25) | (N = 11) | (N = 36) |
| c.2506G>T [p.(Asp836Tyr)]* | | | | |
| Homozygous majority allele (G/G) | 23.70% | 25.80% | 11.80% | 38.70% |
| Minority allele present (G/T & T/T) | 75% | 25% | 0% | 0% |
| c.3140-92T>C** | | | | |
| Homozygous majority allele (T/T) | 23.30% | 25.60% | 12.20% | 38.90% |
| Minority allele present (T/C & C/C) | 57.10% | 28.60% | 0% | 14.30% |

CFTR: cystic fibrosis transmembrane conductance regulator.

* p = 0.024

** p = 0.049.

**Table 4. Demographic, clinical and functional characteristics of asthma with and without airway mucus hypersecretion.**

| Variables | Asthma with hypersecretion | Asthma without hypersecretion | p |
|---|---|---|---|
| | (N = 39) | (N = 59) | |
| Age (y) | 57.43 (11.47) | 49.44 (15.4) | **0.004** |
| Sex (% women) | 61.5% | 49.15% | 0.159 |
| Asthma diagnosis (% adults) | 76.92% | 71.18% | 0.349 |
| BMI (kg/m2) | 27.41 (4.46) | 27.24 (4.93) | 0.864 |
| Severe asthma (%) | 58.97% | 23.72% | **0.005** |
| FEV1/FVC (%) | 64.39 (13.28) | 69.55 (9.61) | **0.041** |
| FeNO (ppb) | 32.45 (25.64) | 39.81 (43.27) | 0.291 |
| Positive bronchodilator test (%) | 23.68% | 35.08% | 0.483 |
| Emergency visits, last 12 months | 2.46 (3.08) | 1.48 (2.24) | 0.074 |
| Oral glucocorticoid treatments, last 12 months | 3.6 (3.7) | 0.86 (1.3) | **0.002** |
| Medium-high oral glucocorticoid doses (%) | 74.35% | 64.4% | 0.678 |
| Rhinitis (%) | 61.53% | 64.44% | 0.360 |
| Polyposis (%) | 20.55% | 8.62% | 0.172 |
| High quality induced sputum (%) | 61% | 27.3% | 0.100 |
| Inflammatory phenotype in induced sputum (%) | paucigranulocytic: 31.25% | paucigranulocytic: 26.32% | 0.596 |
| | neutrophilic: 12.5% | neutrophilic: 26.32% | |
| | eosinophilic: 56.25% | eosinophilic: 47.36% | |
| | (N = 16) | (N = 19) | |
| Positive prick test (%) | 46.15% | 64.4% | 0.216 |
| Blood IgE (IU/mL) | 126.4 (197) | 407.59 (627.6) | **0.003** |
| Absolute eosinophils in peripheral blood (x10E9/L) | 0.39 (0.32) | 0.35 (0.27) | 0.491 |
| Blood polymerase chain reaction (mg/L) | 4.26 (5.57) | 4.20 (6.56) | 0.969 |
| Blood fibrinogen (g/L) | 4.05 (1.02) | 3.98 (0.93) | 0.777 |
| Blood albumin (g/L) | 42.21 (3.06) | 44.14 (3.12) | **0.008** |
| Lymphocytes in induced sputum (%) | 0.71% (0.52) | 1.05% (0.38) | **0.023** |
| ACT <20 (%) | 58.3% | 29.09% | **0.021** |
| AQLQ | 3.55 (2.63) | 2.6 (2.62) | 0.113 |

Values are reported as means (standard deviation) or percentages, as indicated. ACT = Asthma Control Test; AQLQ = Asthma Quality of Life Questionnaire;

BMI = Body mass index; FeNO = fractional exhaled nitric oxide; FEV1 = forced expiratory volume in the first second; FVC = forced vital capacity;

IgE = immunoglobulin E.

10-year follow-up study of 13,756 patients with obstructive lung conditions (chronic bronchitis, emphysema and asthma), Lange et al. [27] observed that airway mucus hypersecretion increased the all-cause mortality risk and that hypersecretion combined with an altered lung function reflected a greater mortality risk for patients with asthma and COPD. Those results would point to the importance of airway mucus hypersecretion in influencing not only the natural course of an obstructive disease but also the associated mortality.

While airway hypersecretion may be an underappreciated condition, clinical practice guidelines for asthma [13] highlight mucus along with bronchoconstriction and inflammation as causes of airway obstruction and airflow limitation. Possible explanations are related to (1) the role played by mucins, (2) the function of toll-like receptors, and (3) the fact of being a carrier of a single-nucleotide polymorphism (SNP) as well as *CFTR* gene mutations. Our findings point to clinical characteristics that differentiated between patients with and without mucus hypersecretion, mainly that hypersecretors were older, had severer asthma, greater bronchial obstruction and poorer asthma control, had lower peripheral blood albumin and lower IgE levels, had lower induced sputum lymphocyte levels, were less likely to have prick test-positive asthma and, finally, needed more short-term oral glucocorticoid treatments in the previous year. Our results corroborate those of Martínez-Rivera et al. [28], who reported that hypersecretion, common in patients with asthma, was associated with greater airway obstruction, poorer asthma control and more exacerbation episodes. Another European prospective respiratory health study followed up over 9 years reported that a determinant of asthma severity was airway mucus hypersecretion [29]. Furthermore, Wheaterall et al. [1], who identified 5 phenotypes, observed that patients with phenotype 5 shared COPD and asthma characteristics (bronchial hyperresponsiveness, elevated FeNO, rhinitis, dermatitis and blood eosinophilia), airway mucus hypersecretion and a poorer response to glucocorticoid anti-inflammatory treatment (resulting in a greater need for treatment and more frequent hospital admissions due to exacerbation episodes).

Chronic airway mucus hypersecretion is classically associated with bronchiectasis and with smoking, although only a minority of smokers develop this symptomatology. A plausible explanation for a greater propensity to hypersecretion may be genetic susceptibility. Dijkstra et al. [30] conducted a genome-wide association study of habitual or former smokers ($\geq$20 pack-years), with (n = 2,704) and without (n = 7,624) chronic mucus hypersecretion, reporting, for all cohorts, a strong association between chronic mucus hypersecretion and SNP rs6577641 located in intron 9 of the special AT-rich sequence-binding protein 1 locus (*SATB1*) gene on chromosome 3 (p = $4.25\times10^{-6}$; OR = 1.17). Likewise reported was that the risk allele was associated with greater mRNA expression of *SATB1* in lung tissue ($4.3\times10^{-9}$) and that airway mucus hypersecretion was associated with greater mRNA expression of *SATB1* in bronchial biopsies from patients with COPD [30]. Another genetic issue raised in the 1990s in relation to airway mucus hypersecretion and asthma was the correlation between carrying a CF mutation an increased asthma risk and a greater deterioration in lung function [3, 4]. Although the asthma-CF carrier relationship is well documented [3], studies carried out between 1998 and 2008 were widely debated, as some failed to find a higher incidence of asthma in CF carriers with the F508del mutation, although they did find a greater deterioration in lung function in asthmatic CF carriers [5, 6]. It was also found that *CFTR* gene mutations altering RNA splicing and/or functional chloride conductance were likely to contribute to the susceptibility and pathogenesis of adult bronchiectasis and pulmonary non-tuberculous mycobacterial infection [31].

To date, of some 1,900 genetic variants described for the *CFRT* gene, that most frequently encountered in the Caucasian population is p.Phe508del (or F508del following the classical nomenclature). Worldwide only 20 mutations occur with a frequency greater than 0.1% [9,

10]. Since the CF inheritance pattern is autosomal recessive, both copies of the mutated *CFTR* gene must be present for the disease to develop, whereas only a single copy is necessary to be a carrier. It is estimated that around 1 in 25–30 Caucasians are carriers of CF mutations–a higher frequency than for other ethnic and racial groups [7, 11, 32]. The fact that the NGS of 97 patients from our sample with asthma confirmed 4% to be CF carriers would tend to corroborate the internal validity of our genetic study. The use of NGS technology is a strength of this study–relative to studies that have typically sequenced just 39–50 *CFTR* gene mutations [3–6, 33–36]–as it permitted sequencing of the entire coding region of the *CFTR* gene along with flanking introns and regions for which pathogenic mutations have been described.

Goodwin et al. [2] reported, for 4 cases of asthma and airway mucus hypersecretion, a neutrophilic inflammatory phenotype accompanied by bronchiectasis, pansinusitis, respiratory infections and mutations and/or polymorphisms in the *CFTR* gene, positing the interesting possibility of an association between mutations and a characteristic phenotype. The results of our NGS study would suggest that, in hypersecretors compared to non-hypersecretors, the NM_000492.3 (CFTR): c.1680-870T>A polymorphism is present to a significantly greater degree: 78.94% vs. 59.32% in the majority allele (thymine nucleotide) and 21.05% vs. 40.67% in the minority allele (adenine nucleotide) (p = 0.036). This polymorphism is described as a SNP variant for CF [37], but given that its role in patients with asthma is still unknown, further studies based on larger samples and specific clinical variables are needed to confirm our findings.

The limitations of this study include the fact that the finding of a single polymorphism can potentially lead to spurious associations, among other reasons, because the variant might be found in linkage disequilibrium with 1 or more other variants and so constitute a characteristic haplotype. Information about haplotypes is becoming increasingly available online, so validating the prognostic or therapeutic utility of this polymorphism would be useful. Other limitations are the lack of a questionnaire that objectively evaluated airway mucus hypersecretion, a possibly unrepresentative sample size and the fact that all the included patients were Caucasian (this may explain the multiple SNPs present in single patient). Strengths include–apart from the above-mentioned use of NGS for the DNA study, with the associated advantages–the inclusion of patients with a verified asthma diagnosis rather than patients from a database–as in previous studies–for whom a confirmed objective diagnosis may not have been available. A final strength is that exhaustive phenotyping was possible because patients underwent extensive clinical, functional and inflammatory testing.

## Conclusions

The findings of this study of patients with asthma and airway mucus hypersecretion suggest the following: (1) a significantly high percentage of patients express the NM_000492.3(CFTR): c.1680-870T>A polymorphism in the *CFTR* gene, and (2) these patients tend to be older, have greater asthma severity and poorer clinical control and have a non-allergic inflammatory phenotype. The findings point to a possibly interesting explanation: the overlap of having asthma and being a carrier of that CF gene mutation may cause a combination of asthma and an attenuated form of CF that would account for the component of bronchial mucosal hypersecretion. More studies are needed to validate the prognostic and therapeutic usefulness of these findings.

## Supporting information

**S1 Appendix. Questionnaires (Spanish and English) used to define bronchial hypersecretion in patients with asthma.**
(ZIP)

**S2 Appendix. Statement regarding sample size representativeness and anonymous data.**
(ZIP)

**S3 Appendix. Tables.**
(ZIP)

## Acknowledgments

The authors would like to thank Ailish M J Maher for translating and reviewing the article. Grateful thanks to the Emerging Asthma Group^, namely, as leaders, Vicente Plaza, Carolina Cisneros and José Serrano (emails: vplaza@santpau.cat; carol9199@yahoo.es; jserrano@separ.es) and as members, Astrid Crespo-Lessmann, Carlos Martínez- Rivera, Nuria Marina, Abel Pallarés-Sanmartín, Silvia Pascual, Juan Luis García-Rivero, Alicia Padilla-Galo and Elena Curto.

## Author Contributions

**Conceptualization:** Astrid Crespo-Lessmann, Carlos Martínez-Rivera, Alicia Padilla-Galo, Elena Curto, Carolina Cisneros, José Serrano, Vicente Plaza.

**Data curation:** Carlos Martínez-Rivera, Nuria Marina, Abel Pallarés-Sanmartín, Silvia Pascual, Juan Luis García-Rivero, Alicia Padilla-Galo.

**Formal analysis:** Astrid Crespo-Lessmann, Silvia Pascual.

**Funding acquisition:** Astrid Crespo-Lessmann, José Serrano.

**Investigation:** Sara Bernal, Ester Rojas, Nuria Marina, Abel Pallarés-Sanmartín, Silvia Pascual.

**Methodology:** Sara Bernal, Elisabeth del Río, Ester Rojas, Alicia Padilla-Galo.

**Project administration:** Juan Luis García-Rivero, Elena Curto, Carolina Cisneros, Montserrat Baiget, Vicente Plaza.

**Supervision:** Carolina Cisneros, José Serrano, Vicente Plaza.

**Validation:** Montserrat Baiget.

**Writing – original draft:** Astrid Crespo-Lessmann.

**Writing – review & editing:** Elena Curto, Vicente Plaza.

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
