## [Decision Letter · Decision Letter 0]

8 Feb 2021

PONE-D-20-37886

Association of the CFTR gene with asthma and airway mucus hypersecretion

PLOS ONE

Dear Dr. Crespo-Lessman,

Thank you for submitting your manuscript to PLOS ONE. After careful consideration, we feel that it has merit but does not fully meet PLOS ONE’s publication criteria as it currently stands. Therefore, we invite you to submit a revised version of the manuscript that addresses the points raised during the review process.

We look forward to receiving your revised manuscript.

Kind regards,

Ritesh Agarwal

Academic Editor

PLOS ONE

Journal Requirements:

2. Please include additional information regarding the survey or questionnaire used in the study and ensure that you have provided sufficient details that others could replicate the analyses.

For instance, if you developed a questionnaire as part of this study and it is not under a copyright more restrictive than CC-BY, please include a copy, in both the original language and English, as Supporting Information.

If the questionnaire is published, please provide a citation to the questionnaire and/or original publication associated with the questionnaire.

3. In your Methods section, please provide additional information about the participant recruitment method and the demographic details of your participants.

Please ensure you have provided sufficient details to replicate the analyses such as:

a) the recruitment date range (month and year),

b) a table of relevant demographic details,

c) a statement as to whether your sample can be considered representative of a larger population, and

d) a description of how participants were recruited.

'I have read the journal's policy and the authors of this manuscript have the following competing interests:

AC has received fees in the last three years for talks at meetings sponsored by Chiesi, Esteve Laboratories, GlaxoSmithKline, Novartis, Ferrer, Zambón and Boehringer Ingelheim, has received travel and attendance expenses for conferences from Novartis, Bial, Teva and FAES Farma and has received funds/grants for research projects from several state agencies and non-profit foundations and from AstraZeneca.

EC reports non-financial support from Astrazeneca, personal fees from Boehringer-Ingleheim, personal fees and non-financial support from Chiesi, non-financial support from Novartis, non-financial support from Menarini, non-financial support from ALK, outside the submitted work.

SB, EdR, ER, CM, NM, AP, SP, JG, AP, CC, JS and MB have no conflicts of interest to declare.

VP has received fees in the last three years for talks at meetings sponsored by AstraZeneca, Boehringer-Ingelheim, MSD and Chiesi, has received travel and attendance expenses for conferences from AstraZeneca, Chiesi and Novartis, has acted as a consultant for ALK, AstraZeneca, Boehringer, MSD, MundiPharma and Sanofi, and has received funds/grants for research projects from several state agencies and non-profit foundations and from AstraZeneca, Chiesi and Menarini.'

a. Please confirm that this does not alter your adherence to all PLOS ONE policies on sharing data and materials, by including the following statement: "This does not alter our adherence to  PLOS ONE policies on sharing data and materials.” (as detailed online in our guide for authors http://journals.plos.org/plosone/s/competing-interests).  If there are restrictions on sharing of data and/or materials, please state these.

Please note that we cannot proceed with consideration of your article until this information has been declared.

5. One of the noted authors is a group:Emerging Asthma Group.

In addition to naming the author group, please list the individual authors and affiliations within this group in the acknowledgments section of your manuscript.

Please also indicate clearly a lead author for this group along with a contact email address.

6. Please include captions for your Supporting Information files at the end of your manuscript, and update any in-text citations to match accordingly. Please see our Supporting Information guidelines for more information: http://journals.plos.org/plosone/s/supporting-information

Additional Editor Comments:

In addition to the reviewer's comments, the authors should also perform workup for allergic bronchopulmonary aspergillosis and sweat chloride test in the study subjects

Reviewers' comments:

Reviewer's Responses to Questions

**Comments to the Author**

1. Is the manuscript technically sound, and do the data support the conclusions?

Reviewer #1: No

Reviewer #2: Yes

2. Has the statistical analysis been performed appropriately and rigorously? 

Reviewer #1: No

Reviewer #2: Yes

3. Have the authors made all data underlying the findings in their manuscript fully available?

Reviewer #1: No

Reviewer #2: Yes

4. Is the manuscript presented in an intelligible fashion and written in standard English?

Reviewer #1: Yes

Reviewer #2: Yes

5. Review Comments to the Author

Reviewer #1: The study investigates the airway mucus hypersecretors and non-hypersecretors with asthma and aims to correlate geneticvariants (mutations or polymorphisms) in the CFTR gene using next-generation sequencing (NGS).

Utilizing pre-defined clinical characteristics the authors undertake a comparative cross-sectional multicentre study including 39 hypersecretors and 61 non-hypersecretors asthma patients

Major comments:

Although the authors follow a thorough protocol for Clinical and demographic data, the statistical methods employed by the authors to study/link the genetic variants in the CFTR gene with the airway mucus secretions are inadequate:

1. The authors use a Multiplicom CFTR panel on illumine MiSeq sequencer to sequence the gene. By calling the SNPs (see minor comments), the authors employ a chi-square test to establish a link between the airway mucus hypersecretors and non-hypersecretors. Although this approach is adequate if one wants to analyze 1 single criterion (here SNPs), this test fails to produce reliable results when multiple SNPs are tested.

As the authors mentioned, they got 50 genetic variants in the CFTR gene. When performing multiple tests, we often encounter the distribution of true positive results to be virtually indistinguishable from the false positive results.

Therefore the authors need to undertake statistical analysis where the significance values that are adjusted for multiple testing. This is needed to control foe the FDR as mentioned above.

It is recommended that the authors re-analyze the SNP data using PLINK software. Care should be taken for issues like population stratification and SNP duplications, SNP strand issues, etc.

Minor Comments:

The authors mention “data interpretation using computerized systems and

databases incorporated in analytical software” on line 201. This statement is highly ambiguous and needs clarification:

1. what is the sequencing depth/coverage?

2. what is panel (horizontal) coverage of the CFTR gene?

3. how were the SNPs called? Mapping to reference? What version of human genome? SNP filtering criteria?

Reviewer #2: Line no. 204 – Please mention the method by which zygosity analysis was performed.

Line no. 223 - It is wriiten that 50 genetic variants were found in CFTR gene. These were found in how many patients. Also mention if any sample had more than 1 variants and mention its implications in the current study.

Line no. 300 - The SNP cant be linked directly to the causation. The samaple size is small to come to this conclusion. The authors can describe it as a plausible association rather than giving it as a definite linkage.

Line no. 374 - No healthy controls were taken for comparison. So, this statement cant be valid.

Line no. 377 - In the supplementary material excel, it was shown than the mulitple SNPs are present in single patient. How do you explain them and what is there significance.

6. PLOS authors have the option to publish the peer review history of their article (what does this mean?). If published, this will include your full peer review and any attached files.

Reviewer #1: No

Reviewer #2: **Yes: **Shivaprakash M Rudramurthy

---

## [Author Response · Author response to Decision Letter 0]

4 Apr 2021

Journal Requirements:

Answer: Done.

2. Please include additional information regarding the survey or questionnaire used in the study and ensure that you have provided sufficient details that others could replicate the analyses.

For instance, if you developed a questionnaire as part of this study and it is not under a copyright more restrictive than CC-BY, please include a copy, in both the original language and English, as Supporting Information.

If the questionnaire is published, please provide a citation to the questionnaire and/or original publication associated with the questionnaire.

Answer: Included patients were administered a questionnaire to evaluate symptoms and type of expectoration. Asthmatic hypersecretion was defined as the presence of cough productive of sputum on most days for at least three months in two successive years.

We have included copies of the questionnaire (in both languages) used to assess bronchial mucus hypersecretion in patients in the appendices. The questionnaire has not been copyrighted.

3. In your Methods section, please provide additional information about the participant recruitment method and the demographic details of your participants.

Please ensure you have provided sufficient details to replicate the analyses such as:

a) the recruitment date range (month and year),

b) a table of relevant demographic details,

c) a statement as to whether your sample can be considered representative of a larger population, and

d) a description of how participants were recruited.

Answer: 

a) The recruitment date range has been included in p.5, methods section, first paragraph.

b) Table 4 shows the demographic, clinical and functional characteristics of asthma with and without airway mucus hypersecretion.

c) It is estimated that 4-5% of the general population are carriers of this disease (CF). To date, some 1,900 genetic variants have been described for the CFRT gene, while only some 20 mutations occur with a frequency greater than 0.1% worldwide (1-2). When sample size was calculated, it was aimed to recruit a total of 100 patients (50 patients for each group), assuming a loss of 10%. This number was calculated using the Granmo V7.10 program, setting the type I error at the usual 5% (α=0.05), for a bilateral approximation and for the minimum difference required for a power of 80% or higher (ß=0.2). However, given the low incidence (not known in fact) of patients with asthma and associated bronchial hypersecretion, it was only possible to recruit 39 patients who met the requirements for asthma with bronchial hypersecretion, and 61 patients with asthma without bronchial hypersecretion, so this study may not be representative of a larger population. A comment on this point has been included in the text (see the discussion section, p.10, third paragraph). We have also included a signed statement with this comment (see Appendix 2). 

d) The patients included in the study, who came from the outpatient clinics of the hospitals participating in the study, were recruited according to the criteria of the participating physicians. A comment on this has been added in the methods section, p.5, first paragraph. 

References:

1. Asencio O, Cobo N, Seculi JL, Casal t, Bosque M. Programa de cribaje neonatal para la fibrosis quística en Cataluña. Investig Clin 2001; 4 (suppl I): 82-3.

2. Javier de Gracia , Antonio Alvarez, Fernando Mata, Luisa Guarner, Montserrat Vendrell, Silvia Gadtner, et al. Cystic Fibrosis in Adults: Study of 111 Patients. Med Clin (Barc) 2002 Nov 9;119(16):605-9.

'I have read the journal's policy and the authors of this manuscript have the following competing interests:

AC has received fees in the last three years for talks at meetings sponsored by Chiesi, Esteve Laboratories, GlaxoSmithKline, Novartis, Ferrer, Zambón and Boehringer Ingelheim, has received travel and attendance expenses for conferences from Novartis, Bial, Teva and FAES Farma and has received funds/grants for research projects from several state agencies and non-profit foundations and from AstraZeneca.

EC reports non-financial support from Astrazeneca, personal fees from Boehringer-Ingleheim, personal fees and non-financial support from Chiesi, non-financial support from Novartis, non-financial support from Menarini, non-financial support from ALK, outside the submitted work.

SB, EdR, ER, CM, NM, AP, SP, JG, AP, CC, JS and MB have no conflicts of interest to declare.

VP has received fees in the last three years for talks at meetings sponsored by AstraZeneca, Boehringer-Ingelheim, MSD and Chiesi, has received travel and attendance expenses for conferences from AstraZeneca, Chiesi and Novartis, has acted as a consultant for ALK, AstraZeneca, Boehringer, MSD, MundiPharma and Sanofi, and has received funds/grants for research projects from several state agencies and non-profit foundations and from AstraZeneca, Chiesi and Menarini.'

a. Please confirm that this does not alter your adherence to all PLOS ONE policies on sharing data and materials, by including the following statement: "This does not alter our adherence to PLOS ONE policies on sharing data and materials.” (as detailed online in our guide for authors http://journals.plos.org/plosone/s/competing-interests). If there are restrictions on sharing of data and/or materials, please state these.

Please note that we cannot proceed with consideration of your article until this information has been declared.

 Answer: See information included on p.12, conflicts of interests section.

 Answer: OK, thank you.

5. One of the noted authors is a group: Emerging Asthma Group.

In addition to naming the author group, please list the individual authors and affiliations within this group in the acknowledgments section of your manuscript.

Please also indicate clearly a lead author for this group along with a contact email address.

 Answer: Done as requested, see Acknowledgements (p.13).

6. Please include captions for your Supporting Information files at the end of your manuscript, and update any in-text citations to match accordingly. Please see our Supporting Information guidelines for more information: http://journals.plos.org/plosone/s/supporting-information

 Answer: Done as requested, see appendices (p.13). 

Additional Editor Comments:

In addition to the reviewer's comments, the authors should also perform workup for allergic bronchopulmonary aspergillosis and sweat chloride test in the study subjects

Answer: All patients with allergic bronchopulmonary aspergillosis were excluded. The sweat chloride test could not be performed, first because we do not have this equipment for adults, and second, because when carrying out a massive sequencing genetic study, if the patient had had cystic fibrosis, this would have been detected. Two comments have been added to the manuscript on these points (p.5, methods section, first and second paragraphs). 

Reviewers' comments:

Reviewer's Responses to Questions

Comments to the Author

1. Is the manuscript technically sound, and do the data support the conclusions?

Reviewer #1: No

Reviewer #2: Yes

2. Has the statistical analysis been performed appropriately and rigorously?

Reviewer #1: No

Reviewer #2: Yes

3. Have the authors made all data underlying the findings in their manuscript fully available?

Reviewer #1: No

Reviewer #2: Yes

4. Is the manuscript presented in an intelligible fashion and written in standard English?

Reviewer #1: Yes

Reviewer #2: Yes

 5. Review Comments to the Author

Reviewer #1: The study investigates the airway mucus hypersecretors and non-hypersecretors with asthma and aims to correlate genetic variants (mutations or polymorphisms) in the CFTR gene using next-generation sequencing (NGS).

Utilizing pre-defined clinical characteristics the authors undertake a comparative cross-sectional multicentre study including 39 hypersecretors and 61 non-hypersecretors asthma patients

Major comments:

Although the authors follow a thorough protocol for Clinical and demographic data, the statistical methods employed by the authors to study/link the genetic variants in the CFTR gene with the airway mucus secretions are inadequate:

1. The authors use a Multiplicom CFTR panel on illumine MiSeq sequencer to sequence the gene. By calling the SNPs (see minor comments), the authors employ a chi-square test to establish a link between the airway mucus hypersecretors and non-hypersecretors. Although this approach is adequate if one wants to analyze 1 single criterion (here SNPs), this test fails to produce reliable results when multiple SNPs are tested.

As the authors mentioned, they got 50 genetic variants in the CFTR gene. When performing multiple tests, we often encounter the distribution of true positive results to be virtually indistinguishable from the false positive results.

Therefore the authors need to undertake statistical analysis where the significance values that are adjusted for multiple testing. This is needed to control foe the FDR as mentioned above.

It is recommended that the authors re-analyze the SNP data using PLINK software. Care should be taken for issues like population stratification and SNP duplications, SNP strand issues, etc.

Answer: We downloaded PLINK to perform the analysis suggested by the reviewer. However, we were unable to use the software because it is a very complex CMD.exe and we have insufficient experience to be able to import, analyse and interpret the results. Therefore, in an attempt to respond to these comments in the manuscript, we have included a paragraph in the manuscript (p.7, statistical analysis section). 

Minor Comments:

The authors mention “data interpretation using computerized systems and

databases incorporated in analytical software” on line 201. This statement is highly ambiguous and needs clarification:

1. what is the sequencing depth/coverage?

2. what is panel (horizontal) coverage of the CFTR gene?

3. how were the SNPs called? Mapping to reference? What version of human genome? SNP filtering criteria?

Answer: We agree with the minor comments made by reviewer #1. For this reason, we have included information on p.6, see “Study of genetic variants in the CFTR gene“, first paragraph. 

Reviewer #2: Line no. 204 – Please mention the method by which zygosity analysis was performed.

Answer: We agree with the comments made by reviewer # 2. We have added a sentence with the requested information in the corresponding paragraph.

Line no. 223 - It is written that 50 genetic variants were found in CFTR gene. These were found in how many patients. Also mention if any sample had more than 1 variants and mention its implications in the current study.

Answer: In Table 1, we have included the frequency of these genetic variants in the majority and minority allele of the studied population and have also modified the first paragraph of the genetic results section (p.7)

Line no. 300 - The SNP cant be linked directly to the causation. The samaple size is small to come to this conclusion. The authors can describe it as a plausible association rather than giving it as a definite linkage.

Answer: Corrected in the discussion (p.8, first paragraph) and in the conclusions (p.12, first paragraph)

Line no. 374 - No healthy controls were taken for comparison. So, this statement cant be valid.

Answer: This statement is not correct, it’s true. 

We have therefore changed the original sentence: “It is estimated that around 4%-5% of the general population are CF carriers, and the fact that the NGS of 98 samples from patients with asthma resulted in 4% being CF carriers demonstrates the internal validity of our genetic study.”

To the following sentence (and have added a new citation (32) in the references section):

“It is estimated that around 1 in 25-30 Caucasians are carriers of CF mutations – a higher frequency than for other ethnic and racial groups (7,11,32). The fact that the NGS of 97 patients from our sample with asthma confirmed 4% to be CF carriers would tend to corroborate the internal validity of our genetic study” (p.10, last paragraph, discussion section).

Line no. 377 - In the supplementary material excel, it was shown than the mulitple SNPs are present in single patient. How do you explain them and what is there significance.

Answer: The multiple SNPs present in a single patient is explained by the fact that not all the patients in the study were Caucasian. This information has been included (p.11, final paragraph, discussion section).

6. PLOS authors have the option to publish the peer review history of their article (what does this mean?). If published, this will include your full peer review and any attached files.

Do you want your identity to be public for this peer review? For information about this choice, including consent withdrawal, please see our Privacy Policy.

Reviewer #1: No

Reviewer #2: Yes: Shivaprakash M Rudramurthy

---

## [Decision Letter · Decision Letter 1]

26 Apr 2021

PONE-D-20-37886R1

Association of the CFTR gene with asthma and airway mucus hypersecretion

PLOS ONE

Dear Dr. Crespo-Lessman,

Thank you for submitting your manuscript to PLOS ONE. After careful consideration, we feel that it has merit but does not fully meet PLOS ONE’s publication criteria as it currently stands. Therefore, we invite you to submit a revised version of the manuscript that addresses the points raised during the review process.

We look forward to receiving your revised manuscript.

Kind regards,

Ritesh Agarwal

Academic Editor

PLOS ONE

Journal Requirements:

Reviewers' comments:

Reviewer's Responses to Questions

**Comments to the Author**

1. If the authors have adequately addressed your comments raised in a previous round of review and you feel that this manuscript is now acceptable for publication, you may indicate that here to bypass the “Comments to the Author” section, enter your conflict of interest statement in the “Confidential to Editor” section, and submit your "Accept" recommendation.

Reviewer #1: All comments have been addressed

Reviewer #2: All comments have been addressed

2. Is the manuscript technically sound, and do the data support the conclusions?

Reviewer #1: Yes

Reviewer #2: Yes

3. Has the statistical analysis been performed appropriately and rigorously? 

Reviewer #1: Yes

Reviewer #2: Yes

4. Have the authors made all data underlying the findings in their manuscript fully available?

Reviewer #1: Yes

Reviewer #2: Yes

5. Is the manuscript presented in an intelligible fashion and written in standard English?

Reviewer #1: Yes

Reviewer #2: Yes

6. Review Comments to the Author

Reviewer #1: Please remove this line from discussion:

"Our study demonstrates that asthma associated with airway mucus hypersecretion is linked to an intronic CFTR gene polymorphism (NM_000492.3(CFTR): c.1680-870T>A)." Unless strong statistical association is established we cannot establish any causality.

Reviewer #2: All comments has been addressed in this revision. The present manuscript is acceptable for publication.

7. PLOS authors have the option to publish the peer review history of their article (what does this mean?). If published, this will include your full peer review and any attached files.

Reviewer #1: No

Reviewer #2: No

---

## [Author Response · Author response to Decision Letter 1]

3 May 2021

Journal Requirements:

Author's reply: All bibliographic citations have been reviewed and their references in the text have been corrected to ensure maximum accuracy between the two of them. All references are cited in the text according to their order of appearance. To our knowledge, none of the mentioned articles has been retracted.

Comments to the Author

1. If the authors have adequately addressed your comments raised in a previous round of review and you feel that this manuscript is now acceptable for publication, you may indicate that here to bypass the “Comments to the Author” section, enter your conflict of interest statement in the “Confidential to Editor” section, and submit your "Accept" recommendation.

Reviewer #1: All comments have been addressed

Reviewer #2: All comments have been addressed

Author: Thank you for your kind answer.

2. Is the manuscript technically sound, and do the data support the conclusions?

Reviewer #1: Yes

Reviewer #2: Yes

Author's reply: Thank you for your kind answer.

3. Has the statistical analysis been performed appropriately and rigorously?

Reviewer #1: Yes

Reviewer #2: Yes

Author's reply: Thank you for your kind answer.

4. Have the authors made all data underlying the findings in their manuscript fully available?

Reviewer #1: Yes

Reviewer #2: Yes

Author's reply: Thank you for your kind answer.

5. Is the manuscript presented in an intelligible fashion and written in standard English?

Reviewer #1: Yes

Reviewer #2: Yes

Author's reply: Thank you for your kind answer.

6. Review Comments to the Author. Please use the space provided to explain your answers to the questions above. You may also include additional comments for the author, including concerns about dual publication, research ethics, or publication ethics. (Please upload your review as an attachment if it exceeds 20,000 characters)

Reviewer #1: Please remove this line from discussion:

"Our study demonstrates that asthma associated with airway mucus hypersecretion is linked to an intronic CFTR gene polymorphism (NM_000492.3(CFTR): c.1680-870T>A)." Unless strong statistical association is established we cannot establish any causality.

Reviewer #2: All comments has been addressed in this revision. The present manuscript is acceptable for publication.

Author's reply to Reviewer #1: We have softened the sentence in the discussion as suggested by reviewer 1. We accept that the study does not have the statistical power to make such a claim, and that it would be necessary to expand the sample. The sentences that follow qualify the context of our findings and the limitations of the sample studied.

Author's reply to Reviewer #2: Thank you for your kind answer.

7. PLOS authors have the option to publish the peer review history of their article (what does this mean?). If published, this will include your full peer review and any attached files.

Do you want your identity to be public for this peer review? For information about this choice, including consent withdrawal, please see our Privacy Policy.

Reviewer #1: No

Reviewer #2: No

---

## [Editor Report · Decision Letter 2]

5 May 2021

Association of the CFTR gene with asthma and airway mucus hypersecretion

PONE-D-20-37886R2

Dear Dr. Crespo-Lessman,

We’re pleased to inform you that your manuscript has been judged scientifically suitable for publication and will be formally accepted for publication once it meets all outstanding technical requirements.

Kind regards,

Ritesh Agarwal

Academic Editor

PLOS ONE
---

## [Editor Report · Acceptance letter]

19 May 2021

PONE-D-20-37886R2 

Association of the *CFTR* gene with asthma and airway mucus hypersecretion 

Dear Dr. Crespo-Lessmann:

I'm pleased to inform you that your manuscript has been deemed suitable for publication in PLOS ONE. Congratulations! Your manuscript is now with our production department. 

Kind regards, 

on behalf of

Dr. Ritesh Agarwal 

Academic Editor

PLOS ONE